# Inhibition of β1-AR/Gαs signaling promotes cardiomyocyte proliferation in juvenile mice through activation of RhoA-YAP axis

**Masahide Sakabe[1,2], Michael Thompson[1,2], Nong Chen[1,2], Mark Verba[1,2], Aishlin Hassan[1,2], Richard Lu[1,2], Mei Xin[1,2]***

[1]Division of Experimental Hematology and Cancer Biology, Cincinnati Children's Hospital Medical Center, Cincinnati, United States; [2]Department of Pediatrics, College of Medicine, University of Cincinnati, Cincinnati, United States

*For correspondence:
mei.xin@cchmc.org

Competing interest: The authors declare that no competing interests exist.

## Abstract

The regeneration potential of the mammalian heart is incredibly limited, as cardiomyocyte proliferation ceases shortly after birth. β-adrenergic receptor (β-AR) blockade has been shown to improve heart functions in response to injury; however, the underlying mechanisms remain poorly understood. Here, we inhibited β-AR signaling in the heart using metoprolol, a cardio-selective β blocker for β1-adrenergic receptor (β1-AR) to examine its role in heart maturation and regeneration in postnatal mice. We found that metoprolol enhanced cardiomyocyte proliferation and promoted cardiac regeneration post myocardial infarction, resulting in reduced scar formation and improved cardiac function. Moreover, the increased cardiomyocyte proliferation was also induced by the genetic deletion of *Gnas*, the gene encoding G protein alpha subunit (Gαs), a downstream effector of β-AR. Genome wide transcriptome analysis revealed that the Hippo-effector YAP, which is associated with immature cardiomyocyte proliferation, was upregulated in the cardiomyocytes of β-blocker treated and *Gnas* cKO hearts. Moreover, the increased YAP activity is modulated by RhoA signaling. Our pharmacological and genetic studies reveal that β1-AR-Gαs-YAP signaling axis is involved in regulating postnatal cardiomyocyte proliferation. These results suggest that inhibiting β-AR-Gαs signaling promotes the regenerative capacity and extends the cardiac regenerative window in juvenile mice by activating YAP-mediated transcriptional programs.

## Editor's evaluation

This manuscript provides strong evidence that β adrenergic signaling regulates cardiomyocyte proliferation in the postnatal period. The authors provide compelling data that inhibition of β adrenergic signaling promotes cardiomyocyte proliferation in juvenile mice through activation of a RhoA-YAP signaling axis.

## Introduction

The capacity to regenerate and repair in response to cardiac injury in the adult mammalian heart is limited. Neonatal mouse hearts retain regenerative potential following cardiac injury up to 7 days after birth *Porrello et al., 2011*; *Porrello et al., 2013*; *Xin et al., 2013b*. Changes after birth such as metabolic state, oxygen level, cardiomyocyte structure and maturity, hormones, and polyploidy are among the factors contributing to the loss of the regenerative potential in the heart *Hirose et al., 2019*; *Vivien et al., 2016*; *Nakada et al., 2017*; *Kimura et al., 2015*; *Siedner et al., 2003*; *Derks*

*and Bergmann, 2020*; *Puente et al., 2014*. For instance, the postnatal metabolic shift from glycolysis to fatty acid oxidation or aerobic respiration-mediated oxidative DNA damage can lead to cardiomyocyte cell cycle arrest postnatally *Nakada et al., 2017*; *Kimura et al., 2015*; *Puente et al., 2014*. In addition, signaling pathways such as Hippo, Neuregulin, ERBB2, Agrin, and thyroid hormone have been shown to regulate cardiac regeneration *Hirose et al., 2019*; *D'Uva et al., 2015*; *Bassat et al., 2017*; *Mahmoud et al., 2015*; *Lin et al., 2014*.

The evolutionarily conserved Hippo signaling pathway is known as a pivotal regulator of organ size and cell proliferation *Zhao et al., 2011*; *Pan, 2010*. It consists of a series of kinases, including MERLIN, MST1/2, and LATS1/2, that phosphorylate the downstream effectors YAP and TAZ, preventing their nuclear translocation and activation of target gene expression *Varelas, 2014*; *Hansen et al., 2015*. Activation of YAP in the embryonic heart, either through the loss of *Mst1/2*, *Sav1*, or forced expression of a constitutively active form of YAP, induces cardiomyocyte proliferation and increases heart size *Xin et al., 2011*; *Heallen et al., 2011*. Moreover, activation of YAP in adult hearts improves cardiac function and reduces scar formation after myocardial infarction (MI) *Xin et al., 2013a*; *Heallen et al., 2013*; *Leach et al., 2017*, suggesting that YAP activation promotes cardiomyocyte regeneration even in the adult mouse heart. However, it remains unknown what upstream signaling cues regulate the Hippo signaling pathway for cardiac regeneration.

The Hippo pathway can be activated by several molecular signals, including G-protein-coupled receptors (GPCRs), cell-cell interactions, and alterations in cytoskeletal dynamics *Yu et al., 2012*; *Wada et al., 2011*; *Kim et al., 2011*. Beta-adrenergic receptors (β-ARs), members of GPCRs that couple to a stimulatory G protein alpha-subunit (Gαs), are essential components of the sympathetic nervous system *Lymperopoulos et al., 2013*. Stimulation of the β-ARs activates Gαs activity leading to an increase in intracellular cAMP levels and the subsequent activation of protein kinase A (PKA), which can result in increased heart rate and contractility *Rockman et al., 2002*; *Molkentin and Dorn, 2001*. Overexpression of Gαs-protein leads to increased myocardial collagen content and fibrosis with variable hypertrophy in mice *Iwase et al., 1997*; *Geng et al., 1999*, suggesting an important role of Gαs in controlling cardiac contractility or hypertrophic response in the heart. Furthermore, activation of Gαs by epinephrine inactivates YAP and inhibits cell growth in various cell lines *Yu et al., 2012*. This presents an opportunity to manipulate cardiomyocyte proliferation through inhibition of the β-AR signaling pathway.

Inhibition of β-AR by β-adrenergic receptor blockade (β-blockers) has been shown to improve survival and symptoms in heart failure patients *Patel and Shaddy, 2010*; *MERIT-HF Study group, 1999*. Gene variants in *GNAS*, which encodes the Gαs protein, are associated with β-blocker-related survival or risk in patients after coronary artery bypass grafting *Frey et al., 2014*. A recent study suggested that a non-selective β-blocker (propranolol) increased the number of cardiomyocytes in neonatal mice *Liu et al., 2019*. Since propranolol inhibits both the β1-AR that is a predominant receptor in the heart muscle and β2-AR that is highly expressed in vascular and non-vascular smooth muscle cells and endothelial cells in non-heart tissues *Cannavo et al., 2013*; *Cannavo and Koch, 2017*; *Flacco et al., 2013*; *Woo and Xiao, 2012*; *Zhu and Steinberg, 2021*; *Wang et al., 2018*, it is not suitable for patients with diabetes or bronchospasm. The 2nd generation β1-blocker, metoprolol, selectively binds to the β1-AR receptor on cardiomyocytes *Ladage et al., 2013*; however, its effect on cardiomyocyte proliferation and heart regeneration have not been explored. Furthermore, the mechanisms by which β-AR-Gαs signaling modulates cardiomyocyte proliferation and heart regeneration remains to be defined.

In this study, we show that treatment with the β1-blocker metoprolol promotes cardiomyocyte proliferation, reduces scar formation, and improves cardiac function after myocardial injury in juvenile mice. Inhibition of the β1-AR downstream effector Gαs activity by genetic deletion of *Gnas* enhances cardiomyocyte proliferation by activating YAP activity through its nuclear localization. Thus, our study demonstrates that β-AR-Gαs signaling represses the regenerative capacity of postnatal cardiomyocytes by inhibiting YAP-activated transcriptional programs. Inhibition of β-AR-Gαs signaling extended the cardiac regenerative window, suggesting a potential therapeutic target for extending the cardiac regeneration window.

## Results

### β1-blocker treatment promotes cardiomyocyte proliferation

β1-AR is the most abundant β-AR subtype present in cardiomyocytes, comprising about 80% of total β-AR. β2-AR, comprising about 20% of β-AR in cardiomyocytes, is highly expressed in vascular and

non-vascular smooth muscle cells *Cannavo et al., 2013*; *Cannavo and Koch, 2017*; *Flacco et al., 2013*; *Woo and Xiao, 2012*; *Zhu and Steinberg, 2021*; *Wang et al., 2018*. To investigate whether blockade β1-AR specifically has a significant effect on cardiomyocyte proliferation and cardiac regeneration, we injected the β1-AR blocker metoprolol (hereinafter, referred to as "β-blocker") into mice for two weeks starting at postnatal day (P) 1 via daily intraperitoneal injection (IP) (*Figure 1a*). The heart rate of β-blocker treated mice was significantly reduced and thickening of the myocardial wall of β-blocker treated hearts was evident at P14 (*Figure 1b and c*). β-blocker treatment induced an increase in the heart weight-to-body weight ratio at P14, but not at P7 (*Figure 1d* and *Figure 1—figure supplement 1a*). Increased proliferating cardiomyocytes in β-blocker-treated hearts at P7 or P14 was confirmed by 5-ethynyl-2'-deoxyuridine (EdU) incorporation, Ki67, PH3, and aurora kinase B (AURKB) immunostaining (*Figure 1e and f*, *Figure 1—figure supplement 1b–e*). We also observed an increased total number of cardiomyocytes and mononucleated cardiomyocytes (a proliferative and regenerative subpopulation of the postnatal heart), in the β-blocker-treated hearts (*Figure 1g* and *Figure 1—figure supplement 1f*). To examine the possibility that the increased heart size is due to cardiac hypertrophy, cardiomyocyte size was measured. No significant difference in cardiomyocyte size was detected between control and β-blocker-treated hearts (*Figure 1—figure supplement 1g and h*), suggesting that the enlarged heart phenotype by the β-blocker treatment is not likely caused by cardiac hypertrophy. Moreover, β-blocker treatment promoted cardiomyocyte proliferation at later time points from P14 to P28, when the majority of cardiomyocytes are matured (*Figure 1h*), suggesting that β1-AR-selective blocker treatment reactivates cell proliferation even in relatively matured cardiomyocytes in vivo.

## β1-blocker treatment extends the cardiac regeneration window after myocardial infarction

To determine whether β-blocker treatment extends the regenerative window in mice, we induced myocardial infarction by permanent ligation of the left anterior descending coronary artery (LAD) at P7 and treated the mice with β-blocker from P8 until P28 (*Figure 2a*). Whereas the vehicle treated mice showed loss of heart tissue, extensive scarring, and dilation post-MI, β-blocker treated mice exhibited significantly reduced left ventricle fibrosis and increased myocardial tissue (*Figure 2b* and *Figure 2—figure supplement 1*). Cell proliferation of cardiomyocytes was upregulated by β-blocker treatment in both the infarct border zone and the remote zone, evidenced by an increase in the number of PH3 positive cardiomyocytes (*Figure 2c–e*), suggesting that β-blocker treatment extends the cardiac regenerative window and enhances cardiomyocyte proliferation in the injured hearts. Moreover, echocardiography analysis indicated that β-blocker treatment led to an improvement in cardiac function post-MI. Both ejection fraction (EF) and fractional shortening (FS) were decreased in all mice relative to sham control mice 1 week after MI, but cardiac function was dramatically enhanced in the β-blocker treated hearts by 3 weeks post-MI (*Figure 2f and g*). To investigate whether the improved cardiac function is due to an acute cardio-protective effect of β-blocker, we measured the infarcted area 1 day post-MI. No significant difference of ischemic area between control and β-blocker-treated hearts was observed (*Figure 2—figure supplement 2*), suggesting that β-blocker didn't provide substantial cardio-protection immediately after MI. These data indicate that β1-selective blocker treatment can extend the cardiac regeneration window and sustain cardiac functions post-MI injury.

The improved cardiac function in the β-blocker treated mice could be due to a larger volume of blood flow into the ventricle, leading to an increase in the force of contraction associated with a slower heart rate. To rule out this possibility, we assessed cardiac function at 4 days after the last β-blocker treatment. Since metoprolol has a short half-life of 3–7 hr, it should be metabolized during these 4 days. Although the heart rates were comparable between β-blocker-treated and saline-treated control mice, we found that cardiac function was still significantly improved in the β-blocker-treated heart (*Figure 2h*). Therefore, these results suggest that the β-blocker treatment improves cardiac function by promoting cardiac regeneration in the injured heart.

## Deletion of *Gnas* promotes cardiomyocyte proliferation

β1-AR is associated with the stimulatory G protein (Gαs) and the activated Gα subunit then regulates the downstream effector molecules such as PKA and cAMP. We hypothesize that blockade of β1-AR promotes cardiomyocyte proliferation through the inhibition of Gαs activity. To test this hypothesis,

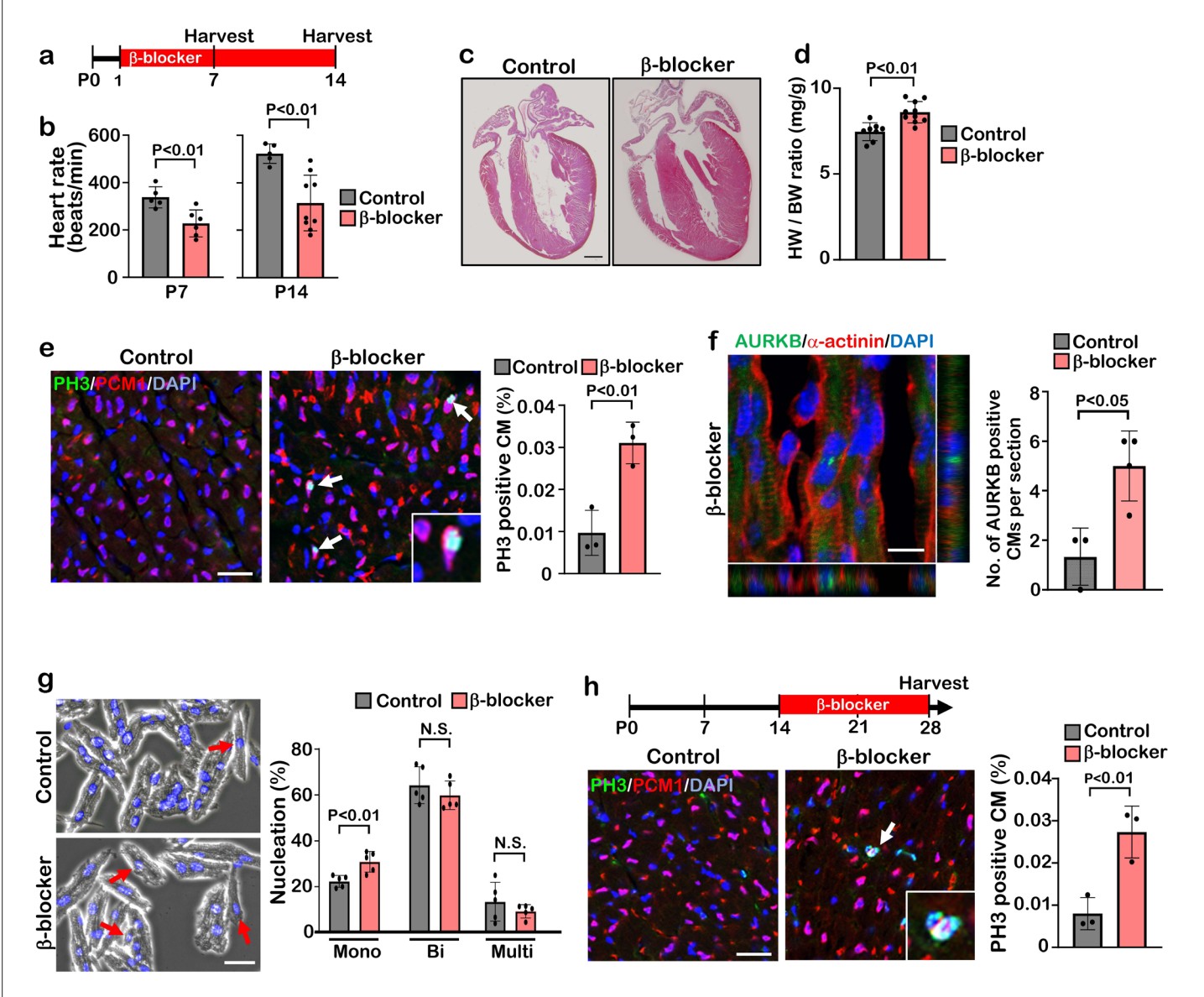

**Figure 1.** β-blocker (Metoprolol) treatment promotes neonatal-juvenile cardiomyocyte proliferation. (**a**) Schematic of experimental timeline. (**b**) Heart rate of the saline (control) and β-blocker-treated mice at P7 and P14. Data are as mean ± SD, Student's t-test, P7, n≥5; P14, n≥5. (**c**) H&E staining of control and β-blocker-treated hearts at P14. Scale bar: 500µm. (**d**) Heart weight (HW) to body weight (BW) ratio at P14. Data are as mean ± SD, Student's t-test, n≥8. (**e**) Co-immunostaining of PH3 and PCM1 of heart sections from control and β-blocker-treated mice at P14 (left panel). Arrows indicate PH3-positive CMs. Inset shows high-magnification image of PH3-positive CM. Quantification of PH3 positive cardiomyocytes (right panel). Data are as mean ± SD, Student's t-test, n=3. Scale bar: 50µm. (**f**) Aurora kinase B (AURKB) staining of P7 β-blocker treated heart. Scale bar: 10 µm (left panel). Quantification of AURKB positive cardiomyocytes at P7 (right panel). Data are as mean ± SD, Student's t-test, n=3. (**g**) DAPI staining of isolated cardiomyocytes from P14 control and β-blocker treated hearts (right panel). Quantification of mono-nucleated, bi-nucleated and multi-nucleated cardiomyocytes in P14 control and β-blocker treated hearts (left panel). Data are as mean ± SD, Student's t-test, n=5. N.S., not significant. Scale bar: 100µm. (**h**) PH3 and PCM1 co-immunostaining of P28 heart sections of control and mice with daily β-blocker treatment from P14 (left panel). Arrow indicates PH3-positive CMs. Inset shows high-magnification image of PH3-positive CM. Quantification of PH3 positive cardiomyocytes (right panel). Data are as mean ± SD, Student's t-test, n=3. Scale bar: 50µm.

The online version of this article includes the following figure supplement(s) for figure 1:

**Figure supplement 1.** β-blocker treatment promotes cardiomyocyte proliferation after the cardiac regeneration window.

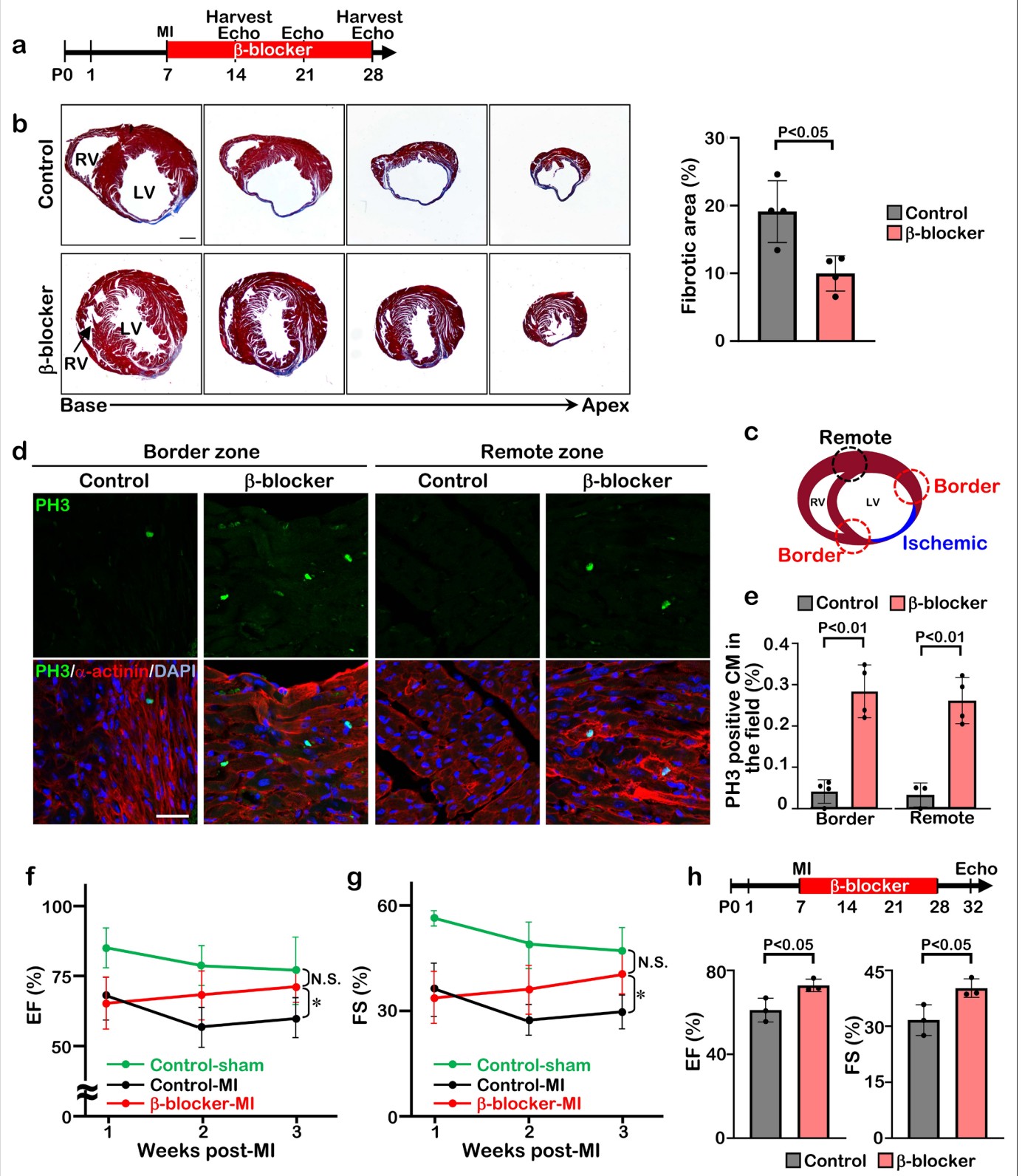

**Figure 2.** β-blocker treatment promotes cardiac regeneration and cardiomyocyte proliferation following injury in juvenile hearts. (**a**) Schematic of experimental timeline. (**b**) Masson's trichrome staining of heart sections from control and β-blocker-treated mice 3 weeks post-MI (left panel). Scale bar: 500μm. Quantification of the fibrotic areas (right panel). Data are as mean ± SD, Student's t-test, n=4. (**c–e**) PH3 and cardiac α-actinin staining of injured hearts treated with saline or β-blocker at 3 weeks post-MI. Data are as mean ± SD, Student's t-test, n≥3. Scale bar: 25μm. (**f–g**) Echocardiographic

*Figure 2 continued on next page*

*Figure 2 continued*

analysis of control and β-blocker-treated mice at 3 weeks post MI surgery. Serial echocardiographic measurements of EF and FS of injured hearts treated with saline (Control) or β-blocker (n=9). ANOVA test, *, p<0.01; N.S., not significant. (**h**) P7 mice were subjected to LAD ligation and treated with β-blocker from P8 to P28. 4 days after final β-blocker treatment, the heart function was assessed by echocardiography. Data are as mean ± SD, Student's t-test, n=3.

The online version of this article includes the following figure supplement(s) for figure 2:

**Figure supplement 1.** Scar area is reduced in β-blocker treated hearts 3 weeks post MI.

**Figure supplement 2.** β-blocker doesn't have cardioprotective effect at juvenile period.

we deleted *Gnas*, the gene encoding Gαs, in the heart by crossing *Gnas^flox/flox* mice with *Myh6^Cre* mouse line. The *Gnas^flox/flox*; *Myh6^Cre* (*Gnas* cKO) hearts did not show any abnormal phenotype compared with littermate controls at birth. However, from P7 onwards, *Gnas* cKO hearts were markedly enlarged and the heart weight-to-body weight ratio was significantly increased (*Figure 3a–b*, *Figure 3—figure supplement 1a*). Furthermore, enzyme-linked immunosorbent assay (ELISA) showed that cAMP levels were significantly reduced in *Gnas* cKO hearts (*Figure 3c*), suggesting that Gαs function was reduced in *Gnas* cKO hearts. Consistent with the downregulation of cAMP level, heart rate was also decreased in *Gnas* cKO (*Figure 3d*).

Immunobiological analysis revealed that the deletion of *Gnas* in the heart resulted in enhanced proliferation of cardiomyocytes as evidenced by the increased number of PH3-positive, and also AURKB-positive cardiomyocytes in *Gnas* cKO hearts (*Figure 3e and f* and *Figure 3—figure supplement 1b*). Further, we dissociated P14 hearts with collagenase and found that the number of cardiomyocytes was significantly increased in *Gnas* cKO hearts (*Figure 3—figure supplement 1c*). No significant difference in cardiomyocyte cell size was detected between the control and *Gnas* cKO hearts (*Figure 3—figure supplement 1d, e*), suggesting that the increased heart size was not due to cardiac hypertrophy. The percentage of mononucleated cardiomyocytes in *Gnas* cKO hearts was indeed higher than that in control hearts (*Figure 3g*). These data suggest that *Gnas* ablation leads to increased cardiomyocyte proliferation and heart size but not cardiac cell hypertrophy.

## Inhibition of β1-AR-Gαs signaling promotes metabolic switch from fatty acid oxidation to glycolysis in cardiomyocytes

To determine the potential mechanisms by which inhibition of β-AR-Gαs promotes cardiomyocyte proliferation and cardiac regeneration, we performed RNA-sequencing analysis (RNA-Seq) using RNA isolated from P7 control, *Gnas* cKO, and β-blocker treated hearts. We identified approximately 2000 differentially regulated genes (fold change ≥1.2) between control vs. *Gnas* cKO and control vs. β-blocker treated hearts. We found that 1076 and 733 genes were down-regulated in *Gnas* cKO and β-blocker treated hearts, respectively, and that 109 genes overlapped between *Gnas* cKO and β-blocker treated hearts (*Figure 4—figure supplement 1a*). Gene ontology (GO) analysis using Enrichr *Chen et al., 2013* indicated that the expression of genes related to fatty acid metabolism, a major source of energy for mature cardiomyocytes, was down-regulated in *Gnas* cKO and β-blocker treated hearts (*Figure 4a*). Gene set enrichment analysis (GSEA) also showed that fatty acid metabolism-related genes were down-regulated in *Gnas* cKO and β-blocker-treated hearts (*Figure 4b*, *Figure 4—figure supplement 1c, d*). In contrast, the expression of genes related to glycolysis, the metabolic pathway utilized by immature cardiomyocytes and associated with cardiomyocyte proliferation, was upregulated in *Gnas* cKO and β-blocker treated hearts (*Figure 4—figure supplement 1c–f*). Quantitative PCR (q-PCR) analysis of fatty acid metabolism-related genes confirmed the RNA-seq results (*Figure 4c*). Moreover, ultrastructural analysis of the heart by electron microscopy (EM) revealed elongated mitochondria in control hearts, whereas mitochondria of *Gnas* cKO hearts were small and round, which is a distinctive phenotype of immature mitochondria, suggesting down-regulation of total energy metabolism in *Gnas* cKO and β-blocker treated hearts. (*Figure 4d*). Consistent with EM images, the copy number of mitochondrial DNA was less in the *Gnas* cKO hearts (*Figure 4e*). These data suggest that cardiomyocytes in the *Gnas* cKO and β-blocker treated hearts exhibit a characteristic feature of immature cardiomyocytes.

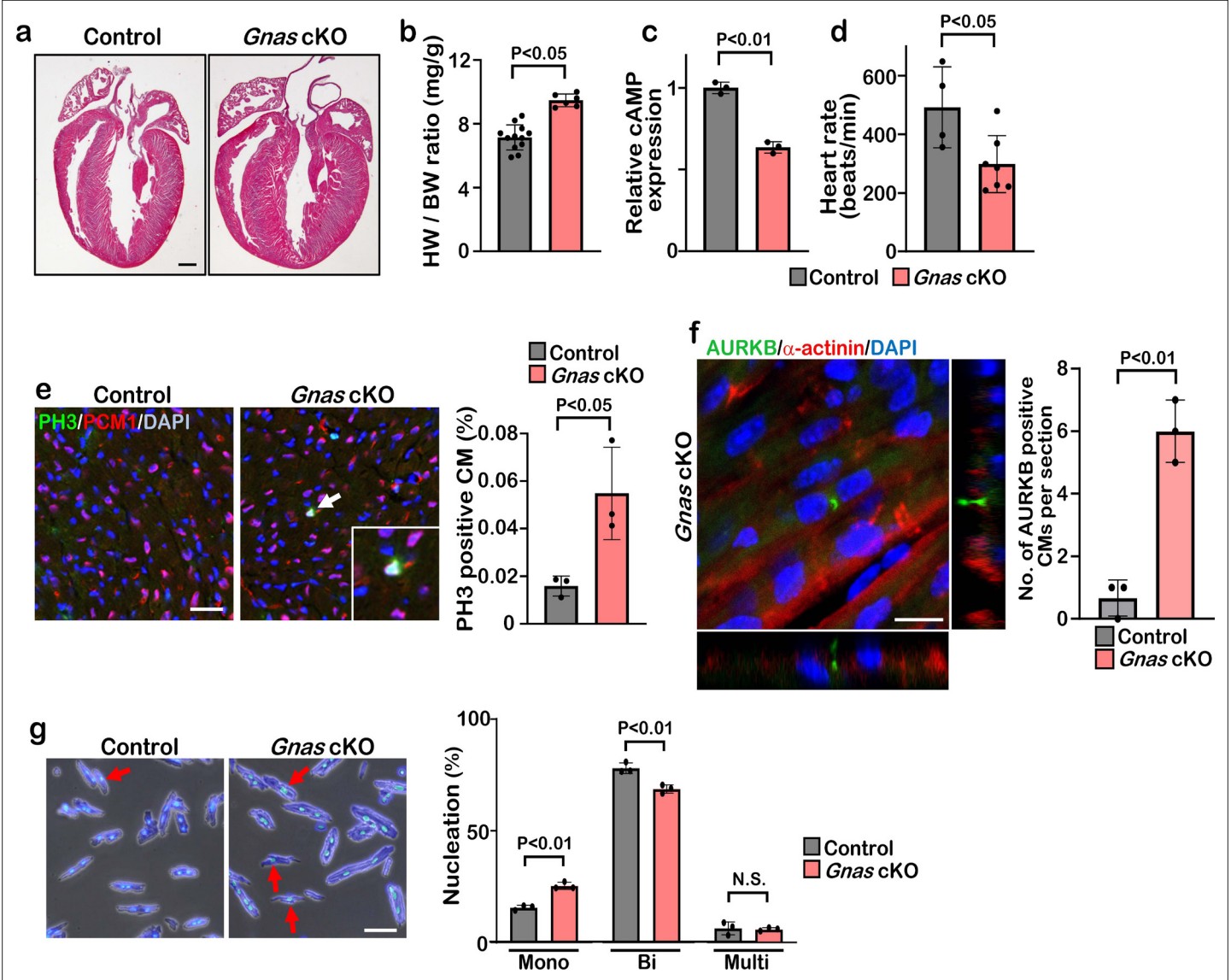

**Figure 3.** Deletion of *Gnas* promotes cardiomyocyte proliferation. (**a**) Hematoxylin and eosin staining of P7 control and *Gnas* cKO heart sections. Scale bar: 500μm. (**b**) Heart weight (HW) to body weight (BW) ratio of P7 *Gnas* cKO mice. Data are as mean ± SD, Student's t-test, n≥5. (**c**) Relative cAMP level in P7 control and *Gnas* cKO hearts. Data are as mean ± SD, Student's t-test, n=3. (**d**) Heart rate at P14 in control and *Gnas* cKO mice. Data are as mean ± SD, Student's t-test, n≥4. (**e**) PH3 and PCM1 staining of left ventricle sections of control and *Gnas* cKO mice at P14 (left panel). Arrow indicates PH3-positive CM. Inset shows high-magnification image of PH3-positive CM. Quantification of PH3 positive cardiomyocytes (right panel). Data are as mean ± SD, Student's t-test, n=3. Scale bar: 50μm. (**f**) Aurora kinase B (AURKB) staining of P7 *Gnas* cKO heart (left panel). Quantification of AURKB-positive cardiomyocytes in the control and *Gnas* cKO mice (right panel). Data are as mean ± SD, Student's t-test, n=3. Scale bar: 10 μm. (**g**) DAPI staining of isolated cardiomyocytes from P14 control and *Gnas* cKO hearts (left panel). Quantification of mono-nucleated, bi-nucleated and multi-nucleated cardiomyocytes in P14 *Gnas* cKO hearts (right panel). Data are as mean ± SD, Student's t-test, n=3. Scale bar: 50μm.

The online version of this article includes the following figure supplement(s) for figure 3:

**Figure supplement 1.** *Gnas* cKO hearts exhibit enlarged phenotype but do not show cardiac hypertrophy.

## Inhibition of β1-AR-Gαs signaling promotes YAP transcriptional activity leading to proliferative immature cardiomyocytes

We found that 1176 and 1845 genes were up-regulated in *Gnas* cKO and β-blocker-treated hearts, respectively, and that 161 overlapping genes identified between *Gnas* cKO and β-blocker-treated hearts (***Figure 4—figure supplement 1b***). GO and GSEA analysis indicated that the expression of genes related to the Hippo signaling pathway was increased in *Gnas* cKO and β-blocker-treated hearts

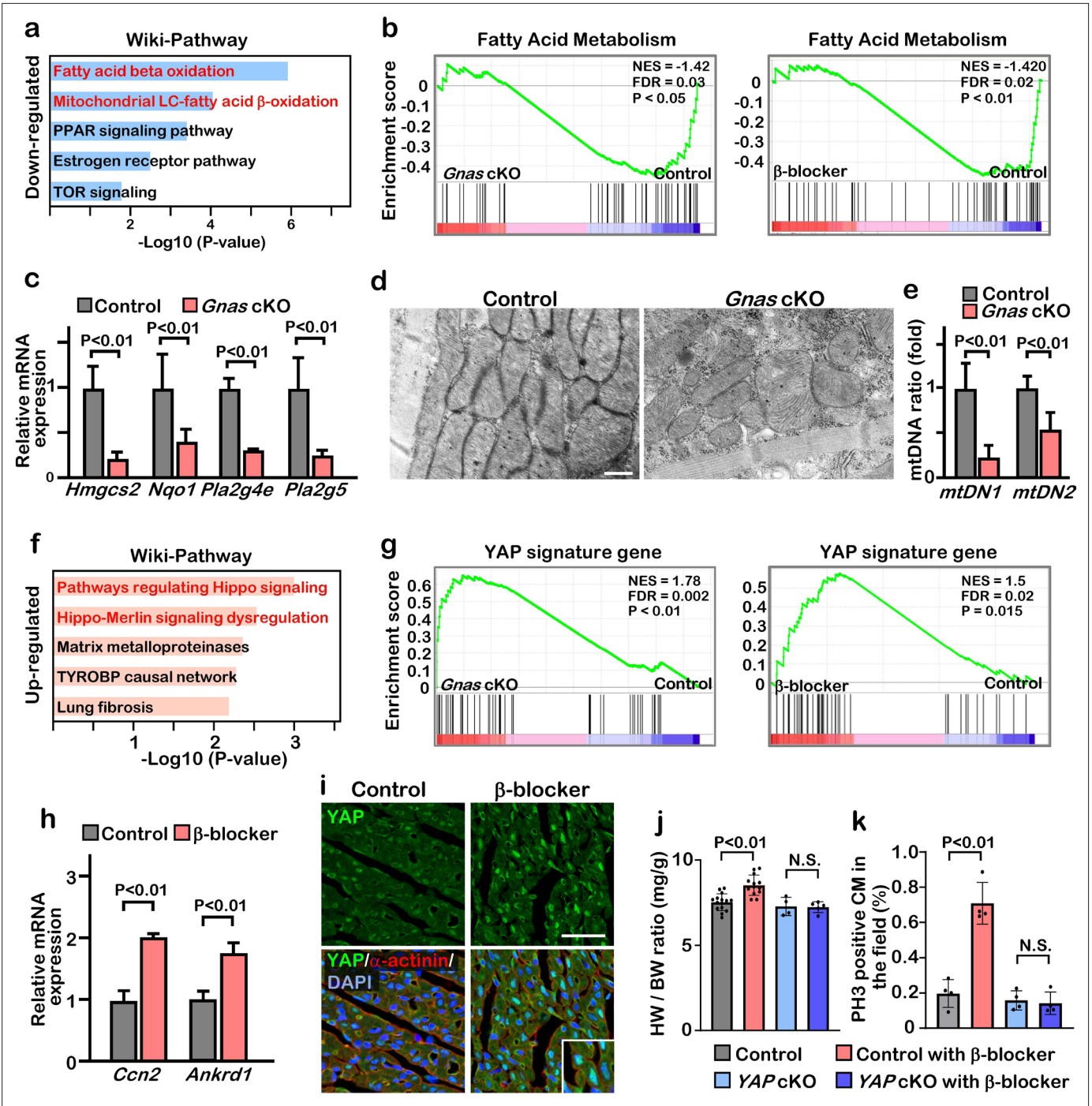

**Figure 4.** Inhibition of βAR-Gαs signaling leads to elevation of YAP activity in the cardiomyocytes. (**a**) Functional enrichment of GO terms for the common down-regulated genes in *Gnas* cKO and β-blocker treated hearts at P7 (fold change ≤0.8). (**b**) GSEA plot shows that fatty acid metabolic genes are down-regulated in *Gnas* cKO and β-blocker-treated hearts. (**c**) Q-PCR analysis of fatty acid metabolism related genes in control and *Gnas* cKO hearts at P14. Data are as mean ± SD, Student's t-test, n=3. (**d**) Transmission electron microscopy images of mitochondria in ventricular cardiomyocytes of P14 control and *Gnas* cKO hearts. Scale bar: 600 nm. (**e**) Q-PCR analysis of mitochondrial DNA in control and *Gnas* cKO hearts at P14. Mitochondrial DNA copy number was normalized to nuclear DNA copy number (mtDN1 vs. H19, and mtDN2 vs. Mx1). Data are as mean ± SD, Student's t-test, n=3. (**f**) Functional enrichment of GO term for the up-regulated genes in *Gnas* cKO and β-blocker treated hearts at P7 (fold change ≥1.2). (**g**) GSEA plot showed that YAP signature genes are up-regulated in *Gnas* cKO and β-blocker treated hearts. (**h**) Q-PCR analysis of YAP target gene expression, *Ccn2* and *Ankrd1*, in control and β-blocker treated hearts at P14. Data are as mean ± SD, Student's t-test, n=3. (**i**) YAP and cardiac α-actinin immunostaining of heart sections from control and β-blocker treated mice at P7 (n=3). Inset shows high-magnification image of nuclear YAP in CMs. Scale bar: 50μm. (**j**) Heart weight (HW) to body weight (BW) ratio of P14 control (n=15), control with β-blocker (n=13), Yap cKO (n=4), and Yap cKO with β-blocker (n=5)

*Figure 4 continued on next page*

*Figure 4 continued*

treated mice. Data are as mean ± SD, ANOVA test, N.S., not significant. (**k**) Quantification of the number of PH3 positive cardiomyocytes per view. Data are as mean ± SD, ANOVA test, n=4. N.S., not significant.

The online version of this article includes the following source data and figure supplement(s) for figure 4:

**Figure supplement 1.** Differential gene expression in *Gnas* cKO and β-blocker-treated hearts.

**Figure supplement 2.** YAP activity is regulated by Gαs.

**Figure supplement 2—source data 1.** Raw data of Western Blots.

(*Figure 4f and g*, *Figure 4—figure supplement 1c, d*). q-PCR analysis confirmed the upregulation of YAP target genes *Ccn2* and *Ankrd1* in *Gnas* cKO and β-blocker treated hearts (*Figure 4h*). YAP cellular localization in cardiomyocytes correlates with heart regenerative capacity. Nuclear YAP is high at P4 when the heart remains regenerative, but low at P7 when the regeneration window is closed. These results are confirmed by western blotting analysis that there is an increase in phospho-YAP (inactive form of YAP) in cardiomyocytes from P4 to P9 mice (*Figure 4—figure supplement 2a, b*). Consistent with upregulated YAP target gene expression, blockade of β1-AR-Gαs signaling by either β-blocker treatment or deletion of *Gnas* promoted retention of YAP nuclear localization in the P7 hearts (*Figure 4i*, *Figure 4—figure supplement 2b*). Fractionation assay also showed that more YAP was detected in the nuclear fraction of P14 β-blocker-treated and *Gnas* cKO hearts compared with littermate controls (*Figure 4—figure supplement 2c, d*). Furthermore, nuclear YAP was detected when the *Gnas* gene was deleted later at P14 upon tamoxifen treatment in the *Gnas-Myh6^{MerCreMer}* cKO mouse line (*Figure 4—figure supplement 2e*). This suggests that Gαs inhibits YAP nuclear localization not only in juvenile but also in young-adult cardiomyocytes. Conversely, when Gαs was activated by epinephrine, an agonist for β-AR, YAP was detected mainly in the cytoplasm at P4 in the control mice; however, YAP remained in the nucleus in the *Gnas* cKO cardiomyocytes when treated with epinephrine (*Figure 4—figure supplement 2f*), suggesting that β-AR signaling inhibits YAP nuclear localization through Gαs activation during cardiomyocyte maturation.

## Ablation of YAP abolished the β-blocker induced cardiomyocyte proliferation

We next investigated whether the increased proliferation phenotype seen with β-blocker treatment was caused by an activation of YAP. We performed β-blocker treatment in the cardiac specific *Yap1* knockout mice *Yap1; Myh6^{Cre}* (*Yap* cKO), which did not show any morphological phenotype in hearts at P14, and found that β-blocker treatment did not increase cardiomyocyte proliferation in the *Yap* cKO hearts (*Figure 4j and k*), suggesting that β-blocker-induced cardiomyocyte proliferation is dependent on YAP functions. Together, our data suggest that Gαs mediates adrenergic signaling to inhibit cardiomyocyte proliferation via inhibition of YAP activity.

## Gαs inhibits cardiomyocyte proliferation through inactivation of the Rho signaling pathway

To gain insight into the molecular mechanisms of how Gαs regulates YAP, we performed pathway analysis based on transcriptomic profiles and found that Rho signaling pathway was activated in *Gnas* cKO and β-blocker-treated hearts (*Figure 5a*). To examine whether Gαs regulates RhoA activity in vivo, we performed active-RhoA pull-down assay using P7 control and *Gnas* cKO hearts. As expected, RhoA activity was increased while the phospho-YAP, the inactive form of YAP (cytoplasmic localized) was dramatically decreased in the *Gnas* cKO heart, compared with littermate controls at P7 (*Figure 5b*).

To confirm this inhibitory effect of Gαs against RhoA, we stimulated Gαs with epinephrine in cultured neonatal rat cardiomyocytes. Epinephrine treatment resulted in a decrease in RhoA activity and an increase in YAP phosphorylation (*Figure 5c*). Moreover, when cardiomyocytes were treated with C3 toxin, a Rho inhibitor, YAP was localized in the cytoplasm, while YAP remained in the nucleus in the saline-treated cardiomyocytes (*Figure 5d*). Treatment with the Rho inhibitors C3 and G04 greatly reduced cardiomyocyte proliferation, as demonstrated by EdU incorporation (*Figure 5e*). Thus, our data suggest that β1-AR-Gαs signaling negatively regulates cardiomyocyte proliferation through the inhibition of RhoA mediated YAP activity. Given that pharmacological inhibition of Gαs promotes

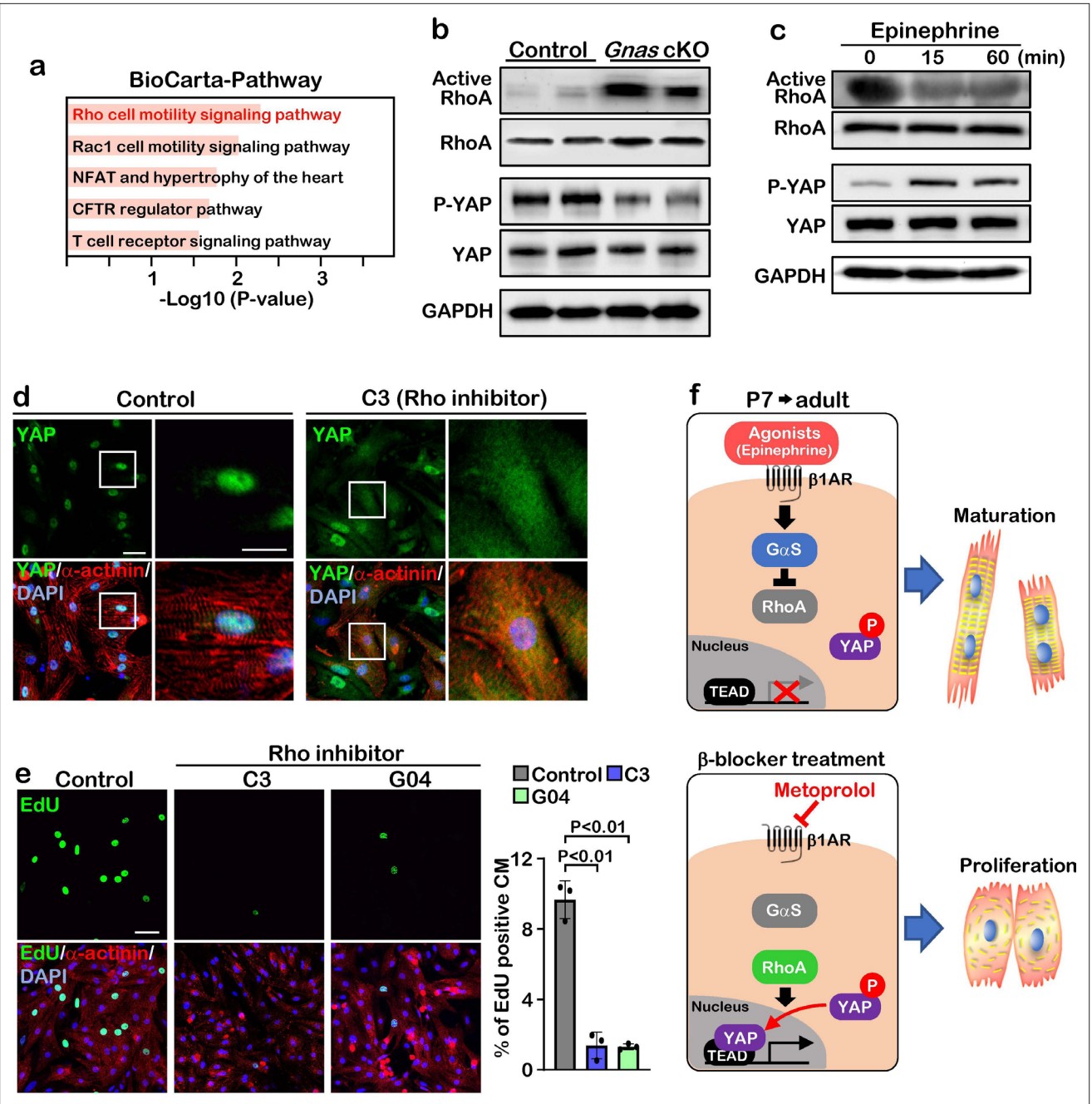

**Figure 5.** GαS regulates cardiomyocyte proliferation through RhoA mediated YAP activation. (**a**) Functional enrichment of GO terms for the common up-regulated genes. (**b**) Active-RhoA pull-down assay and western blot analysis of P7 control and *Gnas* cKO hearts. (**c**) Active-RhoA pull-down assay and western blot analysis of cultured cardiomyocytes with epinephrine treatment. (**d**) Immunostaining of YAP in cultured cardiomyocytes treated with C3 (Rho inhibitor). Scale bar: 50μm (left panel) and 25μm (right panel). (**e**) EdU incorporation assay on rat neonatal cardiomyocytes treated with Rho inhibitors (left panel). Quantification of EdU-labelled proliferating cardiomyocytes stained with cardiac α-actinin. Data are as mean ± SD, n=3 (right panel). Scale bar: 50μm. (**f**) Model of β1-AR-Gαs signaling regulation of cardiomyocyte proliferation. The online version of this article includes the source data for Figure 5.

The online version of this article includes the following source data and figure supplement(s) for figure 5:

**Source data 1.** Raw data of Western Blots.

**Figure supplement 1.** RhoA activity is only detected in the embryonic and postnatal hearts.

**Figure supplement 1—source data 1.** Raw data of Western Blots.

**Figure supplement 2.** Comparison of heart rate at different ages.

cardiomyocyte proliferation through activation of YAP, β-blocker could be used as a potential therapeutic strategy to promote cardiomyocyte proliferation and heart regeneration (*Figure 5f*).

## Discussion

Since the adult mammalian heart has limited potential for re-entry into the cell cycle, cardiomyocyte replenishment after the loss of cardiomyocytes due to myocardial infarction is insufficient to restore heart function *Zhao et al., 2020*; *Cardoso et al., 2020*; *Broughton et al., 2018*. To repair or improve heart function of the injured heart, several strategies using cellular therapies, such as direct cardiac reprogramming, and noncellular therapies have been published *Wang et al., 2021*; *Hashimoto et al., 2018*; *Tzahor and Poss, 2017*; *He et al., 2020*. In this study, we identify β1-adrenergic/Gαs-protein signaling as an inhibitory pathway that restricts the capacity of cardiomyocytes to return to immature proliferative states. When the heart was injured at P7, we show that both pharmacological and genetic inhibition of β1-adrenergic/Gαs-protein reactivate cardiomyocyte proliferation programs and heart regeneration by activating Hippo-YAP signaling, suggesting that β1-adrenergic/Gαs-YAP signaling contributes to extend the cardiac regeneration window at the juvenile stage. At the young-adult age, combination treatment with α/β-blocker and thyroid hormone inhibitor, but not β-blocker itself, is required to enhance cardiomyocyte regeneration after MI surgery *Payumo et al., 2021*, suggesting that additional molecules might be needed to promote adult cardiac regeneration. Further investigation is needed to elucidate the molecular mechanism of how different pathways are involved in promoting adult heart regeneration.

Although neonatal cardiomyocytes have the capacity to proliferate, this regenerative potential is lost in mice one week after birth. During this narrow window, cardiomyocytes undergo one round of DNA synthesis and nuclear division without cytokinesis, which leads to binucleated cardiomyocytes and cell cycle arrest *Li et al., 1996*. At the same time, metabolism in cardiomyocytes switches from glycolysis to fatty acid metabolism, and contractile proteins change from embryonic to neonatal isoforms *Ng et al., 1991*; *Kolwicz et al., 2013*; *Lopaschuk and Jaswal, 2010*. It has been suggested that cardiomyocyte maturation is conversely correlated with its proliferative ability with the increase of DNA content and well aligned contractile structures. Recent studies have also demonstrated that oxidative stress after birth is one driver for cell cycle arrest in neonatal cardiomyocytes *Nakada et al., 2017*; *Kimura et al., 2015*. Despite these findings, the molecular mechanisms of postnatal cardiomyocyte cell cycle withdrawal are not fully understood. In the present study, we found that Gαs signaling activity was correlated with loss of proliferative or regenerative ability of neonatal cardiomyocytes, and inhibition of Gαs activity promotes mono-nucleated cardiomyocyte division and cardiac regeneration through YAP transcriptional activity after the regenerative window. In the canonical Hippo signaling pathway, the major YAP regulators are the LATS1/2 kinases, which phosphorylate and inhibit YAP activity. A previous study reported a model that Rho GTPases inhibit LATS1/2 activity through actin cytoskeleton organization and activate YAP transcriptional activity. Therefore, we consider that Gαs-Rho-cytoskeleton-LATS pathway inhibits YAP and induces loss of proliferative or regenerative ability during cardiomyocyte maturation. Together, inhibition of Gαs induces the de-differentiation of mature cardiomyocytes to an immature state and reactivates YAP transcriptional activity to extend the regeneration window.

It has been shown that cardiac Gαs regulates heart rate and myocardial contractility *Molkentin and Dorn, 2001*; *Tilley and Rockman, 2011*. However, the role of Gαs in cardiomyocytes proliferation is still unclear. Our cardiomyocyte-specific *Gnas* KO mice showed an enlarged heart phenotype with an increase in cardiomyocyte proliferation at the juvenile stage. Conversely, transgenic mice overexpressing Gαs displayed myocardial hypertrophy with increased myocardial collagen content and fibrosis *Iwase et al., 1997*. Together, these results suggest that Gαs plays an important role in maintaining cardiomyocyte homeostasis.

Several signals such as mechanical and oxidative stress have been identified as regulators of the Hippo signaling pathway *Wang et al., 2016b*; *Wang et al., 2016a*; *Lehtinen et al., 2006*; *Geng et al., 2015*. YAP is known as a mechanical sensor due to its ability to alter its localization in response to various environmental stimuli, including alternation of cytoskeletal dynamics and blood flow *Wang et al., 2016b*; *Wang et al., 2016a*; *Foster et al., 2017*; *Vite et al., 2018*. We found that treatment of cultured cardiomyocytes with a Rho inhibitor hinders YAP activity. This is consistent with a previous study showing that the Rho-mediated pathway promotes nuclear localization of YAP in human

embryonic stem cells *Yu et al., 2012*. At present, it is still unclear whether Rho directly regulates YAP activity in cardiomyocytes. A previous report using a cardiomyocyte-specific RhoA transgene revealed heart rate depression and decreased cardiomyocyte contraction *Sah et al., 1999*. We found that RhoA activity was gradually downregulated between P0 and P14 (*Figure 5—figure supplement 1*), and heart rate gradually increased from ~250 beats/min at P0 to ~550 beats/min at P14 (*Sato, 2008*; *Figure 5—figure supplement 2*). During this period, YAP translocates from the nucleus to the cytoplasm, suggesting that RhoA may regulate YAP activity by inhibiting cardiomyocyte contraction rate.

β-AR signaling has been shown to increase cardiac output by enhancing heart rate and contractility via activation of Gαs protein *Salazar et al., 2007*; *Bers, 2002*. β1-AR is expressed in all cardiomyocytes with equal distribution between the left and right ventricles. On the other hand, β2-AR is not only expressed in the cardiomyocytes, but also in vascular and non-vascular smooth muscle cells in multiple tissues. Both β1- and β2-AR are coupled to GαS, while β2-AR can be coupled to Gαi/o as well *Tilley and Rockman, 2011*. Stimulation of β1-AR results in adenylyl cyclase-mediated cAMP generation and activation of PKA, MAP kinase, and calmodulin-dependent protein kinase II (CAMKII) *Bers, 2002*; *Oestreich et al., 2009*; *Mangmool et al., 2010*. Clinical studies have indicated that β-blockers, especially the β1-AR-specific blocker, metoprolol, improve cardiac function and reduce mortality in patients with heart failure and MI *MERIT-HF Study group, 1999*. However, mechanisms underlying the therapeutic effects of β-blockers in heart failure patients are poorly understood. A recent study indicated that in heart failure patients, YAP activity is inactivated by phosphorylation, and that blocking the inhibitory kinase LATS1/2 can reverse heart failure post-MI in mice *Leach et al., 2017*. Similarly, we demonstrate that at the juvenile stage, β-blocker treatment could promote nuclear YAP translocation and cardiac regeneration after MI. Given that YAP activity enhances cardiac regeneration and promotes survival post-MI in our mouse models, activation of Hippo-YAP signaling might be one of the reasons why β-blocker treatment is able to improve heart function in patients.

## Materials and methods

### Mouse experiments

All animal experiments were performed with the approval of the Institutional Animal Care and Use Committee of Cincinnati Children's Hospital Medical Center. Mouse lines harboring the *Gnas* and *Yap* floxed alleles have been described previously *Xin et al., 2011*; *Chen et al., 2005*. The α-Myosin heavy chain (*Myh6*)-Cre (Myh6^Cre) and Myh6^MerCreMer mice were obtained from Jeff Robbins and Jeffery D. Molkentin, respectively *Sanbe et al., 2003*; *Sohal et al., 2001*. Tamoxifen (Sigma) was dissolved in 90% sunflower oil (Sigma)/10% ethanol and stored at –20°C. The tamoxifen solution was injected by intraperitoneal (IP) injection once a day (50 mg/kg). For 5-ethynyl-2-deoxyuridine (EdU) studies, mice were administered an IP injection of EdU (5 µg/kg) once a day. The β-blocker (metoprolol) was dissolved in saline and injected by IP (2 mg/kg) once a day. C57BL/6 J mice were used for the β-blocker treatment studies.

### Myocardial infarction

To induce myocardial infarction (MI) in juvenile mice, we permanently ligated the left anterior descending artery on P7 as previously described *Blom et al., 2016*; *Mahmoud et al., 2014*. Briefly, mice were anaesthetized with isoflurane and the heart was exposed via thoracotomy through the fourth or fifth intercostal space. An 8–0 nylon suture was tied around the left anterior descending coronary artery (LAD). Subsequently, the chest and skin were closed in layers using 6–0 nylon sutures. The mouse was allowed to recover from surgery on a heating pad. Sham-operated mice underwent the same procedure involving anesthesia and thoracotomy without LAD ligation.

### Echocardiography

Assessment of cardiac function on conscious, non-sedated mice was performed with the Vivo 2100 micro-ultrasound system (VisualSonics). Cardiac function and heart rate were measured on M-mode and doppler images.

## Neonatal rat cardiomyocyte isolation and culture

Neonatal rat cardiomyocyte culture was performed using the neonatal cardiomyocyte isolation kit (Cellutron). P2 neonatal cardiomyocytes were plated on tissue culture dishes pre-coated with Sure-Coat (Cellutron) at a density of $2 \times 10^5/cm^2$. After 24 hr, cardiomyocytes were treated with epinephrine (100 μM, Sigma), C3 (1 mg/ml, Cytoskeleton), and GO4 (100 μM, provided by Dr. Yi Zheng).

## Mouse cardiomyocyte isolation

The mouse cardiomyocyte isolation was performed as previously described *Mahmoud et al., 2013*. In brief, P14 hearts were harvested and immediately fixed with 4% paraformaldehyde (PFA) at 4°C overnight. Subsequently, samples were incubated with collagenase B (1.8 mg/ml, Roche) and D (2.4 mg/ml, Roche) for 12 hr at 37°C. The hearts were minced to smaller pieces and the procedure was repeated until no more cardiomyocytes were dissociated from the tissue. The digested cardiomyocytes were stained with 4',6-diamidino-2-phenylindole (DAPI) for nucleation counts. A hemocytometer was used for counting cardiomyocytes.

## Assessment of cardiomyocyte size

The cross-sectional area of cardiomyocytes was assessed using wheat germ agglutinin (WGA) staining. Cryosections were rinsed in PBS and then incubated with WGA conjugated with Alexa Fluor 488 (1:100, Invitrogen). Slides were imaged by Eclipse Ti confocal microscopy with a C2 laser-scanning head (Nikon). ImageJ software (National Institutes of Health) was used to quantify the size of each cell. The area of the digested cardiomyocytes was quantified using ImageJ software based on phase contrast images.

## Histological analysis

Hearts were fixed in 4% PFA at 4°C overnight, embedded in paraffin, and sectioned at 5 μm thickness. Hematoxylin and eosin (H&E) staining was performed following standard protocol. Masson's trichrome and Picrosirius red staining was performed according to standard procedures at CCHMC's pathology core. Fibrotic area was quantified using ImageJ software.

## Immunofluorescence experiments

For immunostaining, hearts were fixed in 4% PFA at 4 °C overnight, embedded in OCT compound (Sakura), and sectioned at 8 μm thickness. For PCM1 staining, we used fresh-frozen (non-fixed) samples and sections were fixed in 10% formalin for 10 min. Sections were blocked with 1% bovine serum albumin (BSA), incubated with primary antibodies against PH3 (rabbit polyclonal, 1:200; Millipore), PCM1 (1:1000; Sigma), cardiac α-actinin (1:200; Sigma), cardiac Troponin T (1:200; Thermo), YAP (1:100; Cell Signaling), and smooth muscle α-actin conjugated with AlexaFluor594 (1:200; Sigma), and were further incubated with Alexa Fluor-conjugated secondary antibodies against mouse or rabbit IgG and with DAPI. For EdU staining, postnatal mice were administered an intraperitoneal (IP) injection of EdU (5 μg/g of mouse body weight) at P5, P6, P12, and P13, and we collected the hearts at P14. EdU incorporation was assessed using Click-IT EdU system (Invitrogen). Fluorescent images were captured using Eclipse Ti confocal microscopy with a C2 laser-scanning head (Nikon).

## RNA sequencing

RNA was extracted from hearts using TRIzol (Invitrogen) followed by purification using RNeasy Mini kit (Qiagen). RNA-seq was performed using two individual animals for control and *Gnas* cKO hearts, or control and β-blocker treated hearts. RNA sequencing was performed by the Center for Medical Genomics, Indiana University School of Medicine. The RNA-seq data generated for this study have been made publicly available via NCBI's GEO (GSE186099). Gene Ontology analysis of gene expression changes was performed using Enrichr *Chen et al., 2013* and Gene Set Enrichment Analysis (GSEA) software.

## Real-time qPCR

Total RNA was isolated using TRIzol according to the manufacture's protocol. cDNA was synthesized from 500 ng of total RNA using PrimeScript RT Master Mix (Takara). Quantitative real-time

PCR (qPCR) was performed using SYBR-Green Master Mix (KAPA) on a StepOnePlus Real-Time PCR system (Applied Biosystems). Values for specific genes were normalized to 18s ribosomal RNA.

## Active RhoA assay

RhoA activity was examined by an effector domain, GST-fusion pull-down protocol, as previously described *Zhu et al., 2000*. Cultured cardiomyocytes or heart tissues were lysed in a lysis buffer containing 1% Triton X-100 and incubated with the glutathione bead-bound GST-Rhotekin. The bead-immobilized GTP-bound RhoA and total RhoA in the lysates were probed by immunoblotting with anti-RhoA antibody (Cell Signaling).

## Western blot analysis

Cultured cardiomyocytes or heart tissues were lysed with 2x sample buffer (BioRad) containing 2-mercaptoethanol and heated for 5 min at 95°C. Equal amounts of protein were run on SDS-polyacrylamide gel and transferred to Immobilon-P membrane (Millipore). Membranes were incubated with anti-YAP (Novus), anti-phospho-YAP (Cell Signaling), and anti-GAPDH (Cell Signaling) antibodies at 4°C overnight. Anti-rabbit horse-radish peroxidase (GE healthcare) was used as the secondary antibody, followed by detection with Super Signal West Pico chemiluminescent substrate (Thermo).

## Fractionation assay

The fractionation assay was performed as previously described *Chen et al., 2009*. For the cytoplasmic extract, hearts were lysed in hypotonic buffer (10 mM HEPES (pH 8), 1.5 mM $MgCl_2$, 10 mM KCl, and 1 mM DTT). Nuclei were then resuspended in hypotonic buffer with 420 mM NaCl, 0.2 mM EDTA, 25% glycerol, 1% NP-40, and 1 mM PMSF.

## Statistical analysis

All datasets were taken from n≥3 biological replicates. Used animal numbers or group numbers are described in the respective figure legends. Animals were genotyped before the experiments and were caged together and treated in the same way. The experiments were not randomized. We calculated p values with unpaired Student's t test or analysis of variance (ANOVA) followed by Tukey-Kramer test with Excel (Microsoft Office). *P*-value < 0.05 was considered to represent a statistically significant difference. Data are presented as mean ± SD.

## Acknowledgements

The authors thank Dr. Jeff Molkentin for insightful discussions and suggestions; Dr. Masayuki Fujii for helpful discussions; Zhifei Xu, Bin Liu, and Hui Sun for technical support; Lingli Xu for Data analysis; Dr. Yi Zheng for the Rho inhibitor GO4; Dr. Eric N Olson for *Yap* floxed mice; Dr. Lee Weinstein for *Gnas* floxed mice. This work was supported by the National Institutes of Health (Grant HL-132211).

## Additional information

### Funding

| Funder | Grant reference number | Author |
|---|---|---|
| National Institutes of Health | HL-132211 | Mei Xin |

The funders had no role in study design, data collection and interpretation, or the decision to submit the work for publication.

### Author contributions

Masahide Sakabe, Conceptualization, Data curation, Formal analysis, Investigation, Visualization, Methodology, Writing – original draft; Michael Thompson, Nong Chen, Mark Verba, Aishlin Hassan, Validation; Richard Lu, Resources, Supervision, Validation, Writing - review and editing; Mei Xin,

Conceptualization, Resources, Formal analysis, Supervision, Funding acquisition, Validation, Investigation, Writing – original draft, Project administration, Writing - review and editing

### Author ORCIDs
Masahide Sakabe ⬛ http://orcid.org/0000-0003-0851-296X
Mei Xin ⬛ http://orcid.org/0000-0002-5732-7501

### Ethics
All animal experiments were performed with the approval of the Institutional Animal Care and Use Committee of Cincinnati Children's Hospital Medical Center (IACUC 2019-0086).

### Decision letter and Author response
Decision letter https://doi.org/10.7554/eLife.74576.sa1
Author response https://doi.org/10.7554/eLife.74576.sa2

## Additional files

### Supplementary files
• Transparent reporting form

### Data availability
RNA seq data have been deposited to GEO under accession code GSE186099.

The following dataset was generated:

| Author(s) | Year | Dataset title | Dataset URL | Database and Identifier |
|---|---|---|---|---|
| Sakabe M, Xin M | 2021 | Gene expression changes in beta-blocker treated neonatal hearts | https://www.ncbi.nlm.nih.gov/geo/query/acc.cgi?&acc=GSE186099 | NCBI Gene Expression Omnibus, GSE186099 |

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

## Appendix 1

### Appendix 1—key resources table

| Reagent type (species) or resource | Designation | Source or reference | Identifiers | Additional information |
|---|---|---|---|---|
| gene (*Mus musculus*) | *Gnas* | NCBI | Gene:14683 | |
| strain, strain background (*Mus musculus*) | C57BL/6 J | The Jackson Laboratory | IMSR_JAX:000664 | |
| genetic reagent (*Mus musculus*) | *Gnas^flox/flox* | The Jackson Laboratory | IMSR_JAX:035239 | |
| genetic reagent (*Mus musculus*) | *Yap^flox/flox* | **Xin et al., 2011** | | |
| genetic reagent (*Mus musculus*) | *Myh6-Cre* | **Sanbe et al., 2003** | | |
| genetic reagent (*Mus musculus*) | *Myh6-MerCreMer* | **Sohal et al., 2001** | | |
| biological sample (*Rattus norvegicus*) | Primary neonatal rat cardiomyocytes | In this paper | | Freshly isolated from neonatal rats |
| antibody | PCM1 (rabbit polyclonal) | Sigma | Car#: HPA023370 | IHC (1:1000) |
| antibody | Phspho-histon H3 (PH3) (mouse monoclonal) | Millipore | Car#: 05–806 | IHC (1:100) |
| antibody | Sarcomeric a-actinin (mouse monoclonal) | Sigma | Car#: A7811 | IHC (1:200) |
| antibody | Cardiac Troponin T (mouse monoclonal) | Thermo | Car#: MA295-P1 | IHC (1:200) |
| antibody | YAP (Rabbit monoclonal) | Cell Signaling | Cat#: 14074 | IHC (1:100) |
| antibody | YAP (Rabbit polyclonal) | Novus | Cat#: NB110-58358 | WB (1:1000) |
| antibody | Phospho-YAP (Rabbit monoclonal) | Cell Signaling | Cat#: 13008 | WB (1:1000) |
| antibody | RhoA (Rabbit monoclonal) | Cell Signaling | Cat#: 2117 | WB (1:1000) |
| antibody | GAPDH (Rabbit monoclonal) | Cell Signaling | Cat#: 2118 | WB (1:1000) |
| antibody | Ki67 (Rabbit monoclonal) | Thermo | Cat#: RM9106 | IHC (1:200) |
| antibody | Aurora B (AurkB) (Rabbit polyclonal) | Abcam | Cat#: ab2254 | IHC (1:100) |
| antibody | Histon H3 (Rabbit polyclonal) | Abcam | Cat#: ab1791 | WB (1:3000) |
| sequence-based reagent | Hmgcs2-F | This paper | qPCR primer | GAAGAGAGCGATGCAGGAAAC |
| sequence-based reagent | Hmgcs2-R | This paper | qPCR primer | GTCCACATATTGGGCTGGAAA |
| sequence-based reagent | Nqo1-F | This paper | qPCR primer | AGGATGGGAGGTACTCGAATC |
| sequence-based reagent | Nqo1-R | This paper | qPCR primer | TGCTAGAGATGACTCGGAAGG |
| sequence-based reagent | Pla2g4e-F | This paper | qPCR primer | AGGTGGAGTTCCTACTCGAAG |

*Appendix 1 Continued on next page*

*Appendix 1 Continued*

| Reagent type (species) or resource | Designation | Source or reference | Identifiers | Additional information |
|---|---|---|---|---|
| sequence-based reagent | Pla2g4e-R | This paper | qPCR primer | TGTTCTCGAAGG AGTCTGTCA |
| sequence-based reagent | Pla2g5-F | This paper | qPCR primer | CCAGGGGGCT TGCTAGAA |
| sequence-based reagent | Pla2g5-R | This paper | qPCR primer | AGCACCAATC AGTGCCATCC |
| sequence-based reagent | mtDN1-F | This paper | qPCR primer | CTCTTATCCACG CTTCCGTTACG |
| sequence-based reagent | mtDN1-R | This paper | qPCR primer | GATGGTGGTAC TCCCGCTGTA |
| sequence-based reagent | mtDN2-F | This paper | qPCR primer | CCCATTCCACT TCTGATTACC |
| sequence-based reagent | mtDN2-R | This paper | qPCR primer | ATGATAGTAGAG TTGAGTAGCG |
| sequence-based reagent | CTGF-F | This paper | qPCR primer | GGGCCTCTT CTGCGATTTC |
| sequence-based reagent | CTGF-R | This paper | qPCR primer | ATCCAGGCAAG TGCATTGGTA |
| sequence-based reagent | Ankrd1-F | This paper | qPCR primer | GGATGTGCCGA GGTTTCTGAA |
| sequence-based reagent | Ankrd1-R | This paper | qPCR primer | GTCCGTTTATAC TCATCGCAGAC |
| commercial assay or kit | Neonatal cardiomyocyte isolation kit | Cellutron | NC-6031 | |
| commercial assay or kit | RNeasy mini kit | Qiagen | 74104 | |
| chemical compound, drug | Epinephrine | Sigma | E4375 | 100 mM |
| chemical compound, drug | C3 | Cytoskeleton | CT04 | 1 mg/ml |
| chemical compound, drug | GO4 | Provided by Dr. Yi Zheng | | 100 mM |
| chemical compound, drug | Tamoxifen | Sigma | T5648 | 50 mg/kg, IP injection |
| chemical compound, drug | Metoprolol (b-blocker) | Sigma | M5391 | 2 mg/kg, IP injection |
| chemical compound, drug | 5-ethynyl-2-deoxyuridine (EdU) | Thermo | Cat#: C10337 | 5 mg/kg, IP injection |
| software, algorithm | ImageJ | National Institutes of Health (NIH) | | |
| other | 4',6-diamidino-2-phenylindole (DAPI) | Invitrogen | D1306 | For nuclear staining |
| other | Wheat germ agglutinin (WGA) | Invitrogen | W11261 | For plasma membrane staining |
| other | Collagenase B | Roche | 11088815001 | 1.8 mg/ml For heart digestion |
| other | Collagenase D | Roche | 11088866001 | 2.4 mg/ml For heart digestion |
| other | 2,3,5-Tripherylterazolium chloride (TTC) | Sigma | T8877 | For staining of ischemic region |

