## [Editor Report]

This manuscript provides strong evidence that β adrenergic signaling regulates cardiomyocyte proliferation in the postnatal period. The authors provide compelling data that inhibition of β adrenergic signaling promotes cardiomyocyte proliferation in juvenile mice through activation of a RhoA-YAP signaling axis.

---

## [Decision Letter]

**Decision letter after peer review:**

Thank you for submitting your article "Inhibition of adrenergic b1-AR/Gas signaling promotes cardiomyocyte proliferation through activation of RhoA-YAP axis" for consideration by *eLife*. Your article has been reviewed by 3 peer reviewers, including Enzo R. Porrello as the Reviewing Editor and Reviewer #1, and the evaluation has been overseen by a Reviewing Editor and Didier Stainier as the Senior Editor.

Essential revisions:

1) Cardio-protection vs regeneration: The authors conclude that their results "suggest that β blockade improves cardiac function by promoting cardiac regeneration in the injured heart". However, effects on CM proliferation are fairly modest (~0.3% pH3-positive CM following MI at P7) and it is unclear whether improvements in cardiac function/fibrosis post-MI are due to induction of CM proliferation. Moreover, the authors have not excluded an acute cardioprotective effect post-MI. Infarct size should be assessed (e.g. using tetrazolium chloride staining) and quantified acutely post-MI (e.g 24 hrs) to distinguish acute cardioprotective effects from bona fide regeneration. If the authors have access to cardiac MRI, this could be used to prove remuscularization has occurred post-MI.

2) Cardiomyocyte proliferation: Additional data are required to demonstrate bona fide cardiomyocyte division following β blockade and Gnas loss-of-function. AurkB staining should be performed to identify cardiomyocytes undergoing cytokinesis, which should be used as a marker for CM division as per other publications in the field. In addition, Supp Figure 1D (quantification of cardiomyocyte number following β blocker treatment) is a critical experiment and should be presented in the main data panels of Figure 1.

3) Given the potential clinical significance and therapeutic implications of the current findings, the authors should determine whether β blockers are sufficient to induce cardiomyocyte cell cycle re-entry and cardiac regeneration following myocardial infarction in adult mice.

4) Additional biochemical evidence is required to demonstrate Yap translocation to the nucleus in Supplemental Figure 5. Immunofluorescence images provided in Supplemental Figures 5 d&f are difficult to interpret. Nuclear translocation of the factor, which is essential for its activation, is not shown here biochemically. On the contrary, the authors show that there is nuclear YAP in basal conditions by immunofluorescence, while this drops to virtually null after epinephrine treatment (bar chart in Suppl. Figure 5f). Incidentally, the over 30% nuclear positivity shown in the graph in basal conditions is difficult to reconcile with the lack of nuclear staining in Suppl. Figure 5b.

5) Please clarify the connection of canonical Hippo signaling and the RhoA effect on Yap. In the current manuscript the mechanistic connection remains somewhat vague. In the original Yu Cell paper from 2021 there is a connection to cytoskeleton. Is that the case here?

6) The authors refer to effects of β blocker treatment on "neonatal heart regeneration". This terminology could be confusing as it is most commonly used to describe experiments in P1 (regenerative) neonatal mice not P7 (non-regenerative) mice. For clarity, it is suggested that the authors rephrase these statements to refer to "prolongation of the regenerative window" in neonatal mice.

7) On several occasions the authors refer to expression of b1-AR specifically in the heart. Is it known whether b1-AR expression is restricted to cardiomyocytes and/or whether it is developmentally regulated from neonatal to adult stages? These data are important with regards to interpretation of systemic metoprolol administration studies. These data should be provided or cited from relevant literature.

8) Β blocker treatment and Gas knockout were both associated with repression of fatty acid oxidation and induction of glycolysis. Are these metabolic transcriptional programs dependent on RhoA/Yap signaling?

9) It would be helpful if the authors showed the overlap between β blocker and Gas KO RNA-seq data sets as a Venn diagram in Figure 4 (or associated Supplementary Figure). Was an FDR cut-off applied to identify DEGs?

10) Figure 4K: data should be presented as %pH3-positive CM as per other figures.

11) Figure 5B: Please replace poor quality Western blot image for Active RhoA.

12) Figure 2f and 2g. Why do EF and FS progressively worsen in the sham animals without MI?

13) Typo: Agrin, not agarin.

14) Please provide dot plots rather than bar graphs.

15) Please clearly state the statistical test used in each figure panel.

*Reviewer #1 (Recommendations for the authors):*

1. Cardio-protection vs regeneration: The authors conclude that their results "suggest that β blockade improves cardiac function by promoting cardiac regeneration in the injured heart". However, effects on CM proliferation are fairly modest (~0.3% pH3-positive CM following MI at P7) and it is unclear whether improvements in cardiac function/fibrosis post-MI are due to induction of CM proliferation. Moreover, the authors have not excluded an acute cardioprotective effect post-MI. Infarct size should be assessed (e.g. using tetrazolium chloride staining) and quantified acutely post-MI (e.g 24 hrs) to distinguish acute cardioprotective effects from bona fide regeneration. In addition, the Discussion should be tempered to account for alternative physiological mechanisms mediating therapeutic effects observed post-MI in addition to regeneration (e.g. immunomodulation, inhibition of cell death, angiogenesis, reduced contractile loading, improved coronary flow, etc).

2. It is unclear why experiments were not performed in adult mice to determine whether β blockade or Gas loss-of-function are sufficient to induce adult cardiomyocyte cell cycle re-entry. Given the potential clinical significance and therapeutic implications of the authors findings, this experiment would significantly strengthen the paper.

3. The authors refer to effects of β blocker treatment on "neonatal heart regeneration". This terminology could be confusing as it is most commonly used to describe experiments in P1 (regenerative) neonatal mice not P7 (non-regenerative) mice. For clarity, it is suggested that the authors rephrase these statements to refer to "prolongation of the regenerative window" in neonatal mice.

4. On several occasions the authors refer to expression of b1-AR specifically in the heart. Is it known whether b1-AR expression is restricted to cardiomyocytes and/or whether it is developmentally regulated from neonatal to adult stages? These data are important with regards to interpretation of systemic metoprolol administration studies. These data should be provided or cited from relevant literature.

5. Β blocker treatment and Gas knockout were both associated with repression of fatty acid oxidation and induction of glycolysis. Are these metabolic transcriptional programs dependent on RhoA/Yap signaling?

Other:

6. Supp Figure 1D (quantification of cardiomyocyte number following β blocker treatment) is a critical experiment and should be presented in the main data panels of Figure 1.

7. It would be helpful if the authors showed the overlap between β blocker and Gas KO RNA-seq data sets as a Venn diagram in Figure 4 (or associated Supplementary Figure). Was an FDR cut-off applied to identify DEGs?

8. Figure 4K: data should be presented as %pH3-positive CM as per other figures.

9. Figure 5B: Please replace poor quality Western blot image for Active RhoA.

*Reviewer #2 (Recommendations for the authors):*

My recommendation to the authors is to strengthen the data on cardiomyocyte proliferation and cardiac regeneration. The use of cardiac MRI to prove remuscularization, should the authors have a microMRI available, the study of BrdU incorporation, the evidence for aurora B localization in midbodies would all strengthen the claim for regeneration.

An additional set of required experiments concerns YAP, as in my earlier text. Providing a mechanism linking betaAR to YAP activation would be also important in this context.

Specific issues.

Figure 2f and 2g. Why do EF and FS progressively worsen in the sham animals without MI?

Agrin, not agarin.

There is no established evidence that Yap translocates from the nucleus to the cytoplasm, quite the reverse. A decrease in the levels of nuclear YAP is the consequence of increased cytoplasmic phosphorylation and degradation.

*Reviewer #3 (Recommendations for the authors):*

The authors study mammalian heart regeneration and study the connection between Yap and β-adrenergic receptor (β-AR) blockade. Interestingly, metoprolol robustly enhanced cardiomyocyte proliferation and promoted cardiac regeneration post myocardial infarction, resulting in reduced scar formation and improved cardiac function. The conclusion was also supported by genetic deletion of Gnas. CMs had an immature cell state with enhanced activity of Hippo-effector YAP. They also find that increased YAP activity is modulated by RhoA.

Comments

1) AurkB should be used as a marker for CM division as per other publications in the field. PHH3 will also mark CMs that are undergoing Karyokinesis without division for example.

2) Please provide dot plots rather than bar graphs.

3) Please clearly state the statistical test used in each figure panel.

4) Figure 5 B Western blot needs to be improved and quantified.

5) Please clarify the connection of canonical Hippo signaling and the RhoA effect on Yap. In the current manuscript the mechanistic connection remains somewhat vague. In the original Yu Cell paper from 2021 there is a connection to cytoskeleton. Is that the case here? Please clarify.

[Editors' note: further revisions were suggested prior to acceptance, as described below.]

Thank you for resubmitting your work entitled "Inhibition of adrenergic β1-AR/Gαs signaling promotes cardiomyocyte proliferation through activation of RhoA-YAP axis" for further consideration by *eLife*. Your revised article has been evaluated by Didier Stainier (Senior Editor) and a Reviewing Editor.

The manuscript has been improved but there are some remaining issues that need to be addressed, as outlined below:

As you will see from the reviewer comments below, there continue to be some concerns about the lack of definitive evidence for cardiomyocyte proliferation and cardiac regeneration in this study. Additional data are required to unequivocally demonstrate that adrenergic β1-AR/Gαs signaling promotes cardiomyocyte proliferation including full progression through cytokinesis. Moreover, as metoprolol is not sufficient to induce adult cardiomyocyte proliferation, the authors' conclusions need to be substantially toned down to reflect the restricted effects on cardiac regeneration in the neonatal period. In addition, reviewer 2's remaining concerns about the biochemical validation of YAP activation should be directly addressed.

*Reviewer #2 (Recommendations for the authors):*

The authors have essentially NOT responded to my comments. First, there is no additional evidence of cardiomyocyte proliferation in this revised version of the manuscript. The images for histone H3 phosphorylation show few positive cells, the identity of which remains debatable. The authors themselves admit that there are so few Aurora B positive cells that not even statistical assessment can be attempted. These findings do not support the statement that "metoprolol robustly enhanced cardiomyocyte proliferation and promoted cardiac regeneration (abstract)". In addition, the new experiment in adult mice (asked by another reviewer) shows that the drug does NOT induce regeneration. Hence, the whole message of the manuscript is quite deceptive.

Second, as far as YAP activation is concerned, I asked for molecular or biochemical evidence of nuclear translocation (for example, by cytoplasmic and nuclear fractionation). This was not provided, while the IF images remain doubtful (cf. my original comments). YAP is known to also regulate cardiomyocyte hypertrophy, which could explain some of the findings presented in this manuscript.

---

## [Author Response]

Essential revisions:1) Cardio-protection vs regeneration: The authors conclude that their results "suggest that β blockade improves cardiac function by promoting cardiac regeneration in the injured heart". However, effects on CM proliferation are fairly modest (~0.3% pH3-positive CM following MI at P7) and it is unclear whether improvements in cardiac function/fibrosis post-MI are due to induction of CM proliferation. Moreover, the authors have not excluded an acute cardioprotective effect post-MI. Infarct size should be assessed (e.g. using tetrazolium chloride staining) and quantified acutely post-MI (e.g 24 hrs) to distinguish acute cardioprotective effects from bona fide regeneration. If the authors have access to cardiac MRI, this could be used to prove remuscularization has occurred post-MI.

We thank the reviewer for pointing out the possibility of an acute cardioprotective effect of b blocker post-MI. As reviewer mentioned that some papers reported the cardioprotective effect of b-blocker, but it is still controversial. To address this question, we performed TTC staining to measure the infarct size 24 hrs post-MI (cardiac MRI is not available in our institute). We observed ischemic area in both saline and b-blocker treated hearts, and there is no significant difference of ischemic area between them, suggesting that the b-blocker (metoprolol) may not have any cardioprotective effects at this neonatal period. We added the data in Supplemental Figure3.

Previous studies regarding neonatal heart regeneration showed that promoting cardiomyocyte proliferation can reduce the infarct size and improve cardiac function post MI. The range of percentage of proliferating CM was 0.1-0.3% (Mahmoud et al., 2013; Nakada et al., 2017; Leach et al., 2017). This range is the same as what we observed in b-blocker-treated MI hearts, it is appropriate to say that β blockade improves cardiac function by promoting cardiac regeneration in the injured heart.

2) Cardiomyocyte proliferation: Additional data are required to demonstrate bona fide cardiomyocyte division following β blockade and Gnas loss-of-function. AurkB staining should be performed to identify cardiomyocytes undergoing cytokinesis, which should be used as a marker for CM division as per other publications in the field. In addition, Supp Figure 1D (quantification of cardiomyocyte number following β blocker treatment) is a critical experiment and should be presented in the main data panels of Figure 1.

We performed AurkB staining using P14 control, b-blocker-treated, and *Gnas* cKO hearts. We didn’t observe any AurkB-positive CMs in the control hearts, whereas b-blocker-treated or *Gnas* cKO hearts showed a few AurkB-positive CMs. Since AurkB-postive CM is rare (one or two AurkB-positive CMs per section), it is hard to perform the statistical analysis. Therefore, we included the AurkB staining images in Supplemental Figure1d and Supplemental Figure 4b. Also, we moved the quantification data of cardiomyocyte number from the original Supplemental Fig1D to Fig1F in the revision.

3) Given the potential clinical significance and therapeutic implications of the current findings, the authors should determine whether β blockers are sufficient to induce cardiomyocyte cell cycle re-entry and cardiac regeneration following myocardial infarction in adult mice.

As the reviewer mentioned, we performed MI surgery using adult mice to investigate whether b-blocker is sufficient to induce cardiac regeneration. We observed huge ischemic area in both the control and b-blocker-treated hearts as shown in Author response image 1. Therefore, in the adult stage, b-blocker treatment may not be sufficient to promote cardiac regeneration. Recent study showed that combination treatment with a/b-blocker and thyroid hormone inhibitor enhanced cardiomyocyte regeneration after MI surgery at P14 when the regeneration window has already closed, suggesting that it may need additional drug to promote cardiac regeneration at adult stage.

**Author response image 1. sa2fig1:** 

4) Additional biochemical evidence is required to demonstrate Yap translocation to the nucleus in Supplemental Figure 5. Immunofluorescence images provided in Supplemental Figures 5 d&f are difficult to interpret. Nuclear translocation of the factor, which is essential for its activation, is not shown here biochemically. On the contrary, the authors show that there is nuclear YAP in basal conditions by immunofluorescence, while this drops to virtually null after epinephrine treatment (bar chart in Suppl. Figure 5f). Incidentally, the over 30% nuclear positivity shown in the graph in basal conditions is difficult to reconcile with the lack of nuclear staining in Suppl. Figure 5b.

In the canonical Hippo signaling pathway, the major YAP regulators are the LATS1/2 kinases, which phosphorylate and inhibit YAP activity. In this paper, we found that loss of Gas function up-regulates RhoA activity in cardiomyocytes. Previous paper (Yu et al., Cell, 2012) reported a model that Rho GTPases inhibit LATS1/2 activity through actin cytoskeleton organization and thereby promote YAP activation. From these results, we consider that Gas-Rho-cytoskeleton-LATS signaling pathway regulates YAP activity in neonatal cardiomyocytes. However, it is still unclear the mechanisms by which actin cytoskeleton controls LATS1/2 activity, therefore further investigation is required to solve the question. We added a Discussion section about the signaling pathway that regulates YAP activity in cardiomyocytes.

We apologize for having an incorrect information regarding the epinephrin treated hearts in the figure legend of Supplemental Figure5. We used “P4 pups” for this experiment instead of “P7 pups” as indicated in the original figure legend. We have made corrections in the new figure legend. As it’s shown in Supplemental Figure6b, YAP localized in the nucleus at P4 when the regeneration window is still open; however, YAP translated to the cytoplasm at P7 when the regeneration window closed. We hypothesized that this cytoplasmic YAP translocation is promoted by b1-AR-Gas signaling pathway. To test our hypothesis, we injected epinephrin into “P4 pups” and we observed cytoplasmic YAP translocation in the WT hearts, but not in the *Gnas* cKO hearts. These data indicate that b1-AR stimulation promotes cytoplasmic YAP localization through Gas. The immunostaining results in Supplemental Figure6c using “P17 hearts” confirms our hypothesis. *Gnas* deletion leads to YAP nuclear localization when YAP has already translocated to the cytoplasm at P17 in the control mice.

Additionally, we performed western blotting analysis using hearts from P4 to P9 and found that phospho-YAP level increased as the heart regeneration window closed. This result correlates with the immunostaining data confirming that YAP nuclear localization correlates with heart regenerative capacity. This piece of data is in the new Supplemental Fig6a.

5) Please clarify the connection of canonical Hippo signaling and the RhoA effect on Yap. In the current manuscript the mechanistic connection remains somewhat vague. In the original Yu Cell paper from 2021 there is a connection to cytoskeleton. Is that the case here?

The canonical Hippo signaling pathway is mediated by MST kinases, LATS kinases, and the downstream transcriptional coactivators YAP and TAZ. As the reviewer mentioned, Yu et al. (Cell, 2012) reported that activation of GPCRs-Rho signaling by Epinephrine inactivates YAP through actin cytoskeleton organization which may controls LATS phosphorylation. In this study, we also found that YAP activity in the neonatal cardiomyocytes is affected by b1AR-Gas-Rho signaling pathway. Moreover, Yu et al. reported that YAP nuclear localization under LPA (Ga12/13 agonist) treatment correlated with levels of cellular actin filaments. As shown in Author response image 2, we also have a piece of data indicating that LPA (Ga12/13 agonist) treatment activates YAP in the wild type P7 hearts, as shown in Author response image 2 (data not included in the manuscript). Therefore, we argue that YAP activity in the cardiomyocyte appears to be regulated by Rho-cytoskeleton-LATS signaling pathway. Since the mechanisms by which actin cytoskeleton organization regulates LATS activity is still unclear, further investigation is required to elucidate it. We added the sentence to clarify the Hippo-RhoA-cytoskeleton-LATS signaling pathway in the discussion.

6) The authors refer to effects of β blocker treatment on "neonatal heart regeneration". This terminology could be confusing as it is most commonly used to describe experiments in P1 (regenerative) neonatal mice not P7 (non-regenerative) mice. For clarity, it is suggested that the authors rephrase these statements to refer to "prolongation of the regenerative window" in neonatal mice.

We totally agreed the reviewer’s comment. We have edited it in the revision.

7) On several occasions the authors refer to expression of b1-AR specifically in the heart. Is it known whether b1-AR expression is restricted to cardiomyocytes and/or whether it is developmentally regulated from neonatal to adult stages? These data are important with regards to interpretation of systemic metoprolol administration studies. These data should be provided or cited from relevant literature.

It is well known that b1-AR is the predominant bAR subtype in cardiac muscle comprising 80% of bARs in cardiomyocytes. It is also expressed in sinoatrial node, and atrioventricular node, but juxtaglomerular cell and adipose tissue. Since b1-AR knockout mice die prenatally between E10.5 and E18.5, b1-AR may have a critical role for the adrenergic system during embryonic heart development. However, previous paper reported that b1-AR mRNA expression in adult hearts significantly increased by fivefold than E11.5 hearts. Moreover, basal cAMP level was significantly increased in the developed hearts (*Feridooni et al., 2017, Am. J. Physiol. Heart Circ. Physiol.*). Therefore, these results suggest that b1-AR expression and stimulation is developmentally regulated from neonatal to adult stage. These data also support our model that inhibition of b1-AR signaling by b-blocker extended the cardiac regenerative window. We discussed b1-AR expression and stimulation during heart development in the discussion part.

8) Β blocker treatment and Gas knockout were both associated with repression of fatty acid oxidation and induction of glycolysis. Are these metabolic transcriptional programs dependent on RhoA/Yap signaling?

There are pieces of evidence indicating that in several cancer cell lines, YAP is involved in metabolism regulation to promote glycolysis (*Koo and Guan, 2018, Cell Metabolism*). Other studies also reported that cells having constitutively active YAP promoted glucose utilization, indicating an increase in glycolysis. In fact, YAP promotes transcription of glycolysis related genes, such as GLUT3 or HK2, by interacting with TEAD. In the ischemic hearts, upregulation of YAP occurs one week after TAC, and then we can see increased glucose uptake in adult cardiomyocytes (*Kashihara and Sadoshima, 2019, J. Cardiovasc. Pharmacol.*). Our transcriptome RNA seq data also suggest that YAP may promotes glycolysis in cardiomyocytes in response to loss of Gas function.

9) It would be helpful if the authors showed the overlap between β blocker and Gas KO RNA-seq data sets as a Venn diagram in Figure 4 (or associated Supplementary Figure). Was an FDR cut-off applied to identify DEGs?

We added the Venn diagram of down-regulated and up-regulated genes in *Gnas* cKO and b-blocker treated hearts in Supplemental Figure 5a and 5b. We used FPKM≥0.5 and |fold change|≥1.2 (not FDR cut-off) to identify the differential gene expression. We used FDR for GSEA analysis.

10) Figure 4K: data should be presented as %pH3-positive CM as per other figures.

We revised the data, following the reviewer’s suggestion.

11) Figure 5B: Please replace poor quality Western blot image for Active RhoA.

We have replaced the Western blot image for active RhoA in Figure 5B.

12) Figure 2f and 2g. Why do EF and FS progressively worsen in the sham animals without MI?

As reviewer mentioned, we can see that EF and FS of sham animals appear to be getting worse when we compared 1 week (P14) and 3 weeks (P28) post-MI data. It has been previously reported that the neonatal EF and FS are slightly higher than juvenile mice (P10 vs. P35) (Wiesmann et al., *Am.J.Physiol.Heart Circ.Physiol.*, 2000). Therefore, we consider that the reduction of EF and FS in the sham animals what we observed could be normal phenomenon as shown by other group for the control mice.

13) Typo: Agrin, not agarin.

We have corrected the typo (Page3, Line8).

14) Please provide dot plots rather than bar graphs.

We revised all graphs from bar graph to dot plot.

15) Please clearly state the statistical test used in each figure panel.

We added the types of statistical method in each figure legends.

[Editors' note: further revisions were suggested prior to acceptance, as described below.]

The manuscript has been improved but there are some remaining issues that need to be addressed, as outlined below:As you will see from the reviewer comments below, there continue to be some concerns about the lack of definitive evidence for cardiomyocyte proliferation and cardiac regeneration in this study. Additional data are required to unequivocally demonstrate that adrenergic β1-AR/Gαs signaling promotes cardiomyocyte proliferation including full progression through cytokinesis. Moreover, as metoprolol is not sufficient to induce adult cardiomyocyte proliferation, the authors' conclusions need to be substantially toned down to reflect the restricted effects on cardiac regeneration in the neonatal period. In addition, reviewer 2's remaining concerns about the biochemical validation of YAP activation should be directly addressed.

We thank the reviewers of their positive feedback about our revised manuscript. We are sorry that some concerns regarding cardiomyocyte proliferation and YAP activation still remain not fully addressed. In this resubmission, we have provided EdU and Aurora B Kinase immunostaining data to demonstrate that β1-AR/Gαs signaling promotes cardiomyocyte proliferation from DNA synthesis (EdU) to nuclear division (Phospho- Histone H3) eventually to cytokinesis (Aurora B). We further validated YAP activation via nuclear translocation by Fractionation Assay. Please see the detailed responses to reviewer 2’s concerns below.

We also toned down the conclusions to reflect the restricted cardiac regeneration effect in the juvenile stage (in the revised abstract, text and discussion). The changes are marked in the text.

Reviewer #2 (Recommendations for the authors):The authors have essentially NOT responded to my comments. First, there is no additional evidence of cardiomyocyte proliferation in this revised version of the manuscript. The images for histone H3 phosphorylation show few positive cells, the identity of which remains debatable. The authors themselves admit that there are so few Aurora B positive cells that not even statistical assessment can be attempted. These findings do not support the statement that "metoprolol robustly enhanced cardiomyocyte proliferation and promoted cardiac regeneration (abstract)". In addition, the new experiment in adult mice (asked by another reviewer) shows that the drug does NOT induce regeneration. Hence, the whole message of the manuscript is quite deceptive.

We appreciate the reviewer’s comments. To show the proliferating cardiomyocytes more clearly, in this new revision, we included insets of higher magnification in the Phospho-histone H3 staining images in New Figures (Figure 1e and h and Figure 3e).

To show the full spectrum of proliferating cardiomyocytes, we performed EdU incorporation assay in the *β-*blocker treated hearts. A significant increase in the number of EdU positive and PCM1 (cardiomyocyte nuclear marker) positive cells was detected in the *β-*blocker treated hearts at P14, suggesting an increase in DNA synthesis (New Figure 1—figure supplement 1d).

In the last revision, we did observe Aurora B positive cardiomyocytes in *β-*blocker treated and *Gnas* cKO hearts at P14, but did not detect any Aurora B positive cells in the control hearts confirming that cardiomyocytes ceased proliferation after the regeneration window in the control but ablation of *Gnas* indeed improved cytokinesis. In this new revision, we provided statistical analysis of Aurora B staining of P14 hearts (New Figure 1—figure supplement 1e) as well as new Aurora B data of *β-*blocker treated and *Gnas* cKO hearts at P7. As shown in the new Figure 1f, 3f, Figure 1—figure supplement 1e and Figure 3—figure supplement 1b the number of Aurora B positive cardiomyocytes was significantly increased in both *β-*blocker treated hearts and *Gnas* cKO hearts at P7 and P14. These new results suggest that inhibition of β1-AR/Gαs signaling promotes cardiomyocyte proliferation at the juvenile stage. The texts were edited to reflect these changes (New Page4; line22-23 and Page7; line14).

We have toned down our conclusions regarding the effect of b-blocker on cardiac regeneration strictly to juvenile stage as it’s not sufficiently to induce adult heart regeneration. Please see the new abstract (page1), text (Page4; line1) and discussion (Page12; line5-13, Page13; line16, and Page14; line22). Our study suggests that inhibition of β1-AR/Gαs contributes to extending cardiac regeneration at the juvenile stage. Other pathways may be needed to promote adult cardiomyocyte proliferation and cardiac regeneration, which will need further investigation.

Second, as far as YAP activation is concerned, I asked for molecular or biochemical evidence of nuclear translocation (for example, by cytoplasmic and nuclear fractionation). This was not provided, while the IF images remain doubtful (cf. my original comments). YAP is known to also regulate cardiomyocyte hypertrophy, which could explain some of the findings presented in this manuscript.

We appreciate the reviewer’s comments and have a better understanding of the experiment that the reviewer suggested. As requested, we performed fractionation assay using P14 β1- blocker treated and *Gnas* cKO hearts. YAP is present in the nuclear fraction in both β1- blocker treated and *Gnas* cKO, but barely detectable in the control hearts. Cytoplasmic YAP is comparable with the controls. The nuclear fraction and cytoplasmic fraction were confirmed by Histone H3 and GAPDH, respectively. These data support the immunostaining results of increased YAP nuclear localization in both β1- blocker treated and *Gnas* cKO hearts. We have included these new data in Figure 4—figure supplement 2c and 2d and clarified these points in the manuscript (Page9; line19-21). Also, we included an inset with higher magnification to show nuclear YAP in the cardiomyocytes (new Figure 4—figure supplement 2b and e).

We appreciate that the reviewer pointed out that YAP may regulate cardiomyocyte hypertrophy, which was also one of our concerns. We used two independent methods (WGA and isolated cardiomyocytes) and showed in our last revision that no significant difference in cardiomyocyte size was observed between control and b-blocker treated or Gnas cKO hearts. The data are now in Figure 1—figure supplement 1g and h and Figure 3—figure supplement 1d and e. Therefore, the enlarged heart phenotype is not likely due to cardiac hypertrophy (Page5 line5-10 and Page7; line 18-20).